# PolarBearVidID: A Video-Based Re-Identification Benchmark Dataset for Polar Bears

**DOI:** 10.3390/ani13050801

**Published:** 2023-02-23

**Authors:** Matthias Zuerl, Richard Dirauf, Franz Koeferl, Nils Steinlein, Jonas Sueskind, Dario Zanca, Ingrid Brehm, Lorenzo von Fersen, Bjoern Eskofier

**Affiliations:** 1Machine Learning and Data Analytics Lab, Department Artificial Intelligence in Biomedical Engineering, Friedrich-Alexander-Universität Erlangen-Nürnberg, 91052 Erlangen, Germany; 2Animal Physiology, Department Biology, Friedrich-Alexander-Universität Erlangen-Nürnberg, 91058 Erlangen, Germany; 3Nuremberg Zoo, 90480 Nuremberg, Germany

**Keywords:** deep learning, re-identification, computer vision, animal identification, animal welfare, automated behavior analysis, motion features, video-based method, dataset

## Abstract

**Simple Summary:**

Zoos use automated systems to study animal behavior. These systems need to be able to identify animals from different cameras. This can be challenging, as individuals of the same species might look very alike. AI is the best way to automatically perform this task, especially when using videos instead of images because they show the animal’s movement as additional information. To train the AI model, one needs to have data. This study introduces a new dataset called *PolarBearVidID* that includes video sequences of 13 polar bears in various poses and lighting conditions. Our AI model is able to identify them with 96.6% accuracy. This shows that using the animals’ movements can help identify them.

**Abstract:**

Automated monitoring systems have become increasingly important for zoological institutions in the study of their animals’ behavior. One crucial processing step for such a system is the re-identification of individuals when using multiple cameras. Deep learning approaches have become the standard methodology for this task. Especially video-based methods promise to achieve a good performance in re-identification, as they can leverage the movement of an animal as an additional feature. This is especially important for applications in zoos, where one has to overcome specific challenges such as changing lighting conditions, occlusions or low image resolutions. However, large amounts of labeled data are needed to train such a deep learning model. We provide an extensively annotated dataset including 13 individual polar bears shown in 1431 sequences, which is an equivalent of 138,363 images. *PolarBearVidID* is the first video-based re-identification dataset for a non-human species to date. Unlike typical human benchmark re-identification datasets, the polar bears were filmed in a range of unconstrained poses and lighting conditions. Additionally, a video-based re-identification approach is trained and tested on this dataset. The results show that the animals can be identified with a rank-1 accuracy of 96.6%. We thereby show that the movement of individual animals is a characteristic feature and it can be utilized for re-identification.

## 1. Introduction

Assessing animal welfare is a major challenge for every animal-keeping institution and thus focus of biological research. It is not trivial to obtain a comprehensive picture of an animal’s physical and psychological condition [1]. One crucial tool for investigating animal welfare is the direct observation of the animals under care. Compared to other approaches, such as hormone level evaluation or blood analysis, behavioral observation offers the advantage that it is an inexpensive and non-invasive method [2,3,4]. Nevertheless, it is common for many biologists, veterinarians, and animal caretakers to observe the animals manually, which comes with severe limitations. It is very labor-intensive, time-consuming and cannot be carried out continuously. Moreover, it is prone to human mistakes. Automated camera-based observation systems help to solve these issues and allow for continuous recording and analysis. Unlike a human observer, such systems can record multiple animals simultaneously on a 24/7 scale. The development of camera-based observation systems has increasingly been carried out in the last few years as advances in deep learning continue to offer new tools to study animal ecology and behavior [5,6,7,8,9].

The most crucial and challenging step for such a system is the identification of individual animals in the recorded data. Only by performing this stage can the behavior of individuals be analyzed. This identification step helps biologists and animal keepers to tailor possible measures or treatments to individual needs. The problem of identifying animals in different camera views is called re-identification (short re-ID).

Re-ID refers to the task of ranking a list of known individuals (the *gallery*) when confronted with a new image or video (the *query sample*). The generated list contains the best-matching identities in descending order. Ideally, the first listed individual corresponds to the animal shown in the query sample [10]. Researchers initially focused on re-identification in humans. Large benchmark datasets such as *Mars* [11] or *Market-1501* [12] enabled the development of various methods for this task. In recent years, some of these approaches could be utilized for re-ID in animals. However, there are still numerous gaps in this area of research, which we are helping to fill through our work.

Deep learning approaches to identify animals were first introduced in 2014 and since then, the number of methods used for animal re-ID is increasing [13]. Many of them tackle re-ID for specific species with unique visual features such as salamanders [14], manta rays [15], cows [16] or Amur tigers [17]. In recent years, species-unspecific approaches have been introduced more and more frequently. These methods offer the advantage of not needing unique visual markers such as fur stripes or skin patterns. Most well-performing approaches are based on convolutional neural networks (CNNs). Freytag et al. [18] showed that the CNN architecture AlexNET outperformed all previous re-ID approaches on two groups of chimpanzees. Brust et al. [19] followed the same approach and used AlexNET for the re-ID of gorillas. Schneider et al. [20] took a different path and showed that similarity learning networks could be utilised to re-identify animals across species without handcrafted feature extraction. This approach is promising as it performed well on five different species: humans, chimpanzees, whales, fruit flies and tigers. In a few projects, animal re-ID methods are already embedded in frameworks to collect information about single individuals’ behavior automatically. Marks et al. [6] present an end-to-end pipeline to extract specific behaviors (e.g., social grooming or object interaction) as well as the pose of the animals. Their approach is species-unspecific and was tested on primates and mice, showing promising results. In our previous work [5], we proposed a similar end-to-end framework with a focus on fast inference times to analyse individual polar bears’ trajectories over long time periods.

In automated observation of animal behavior, videos are the primary data source. However, all previously listed methods for re-ID in animals are image-based approaches. Nevertheless, the movement and gait of humans are individual characteristic features used for re-ID in several methods [21]. Video-based approaches aim at incorporating these individual movement characteristics by embedding not only spatial, but also temporal information into their feature representation. When conducting observations of animals in zoological facilities, it is common to encounter low-resolution videos with occlusions. In such cases, utilizing all available information from the video, rather than relying solely on a single image, is essential for successful animal re-identification. In the future, we expect video-based approaches to be used more and more extensively for automated observation of animals in zoos. At this point, however, there are no video-based approaches for re-ID in animals.

Developing deep learning-based approaches for re-ID in animals requires large amounts of labeled data. Unlike in humans, it is difficult to record a large number of individual animals. In particular, publicly available datasets are very limited [20]. Table 1 shows a list of publicly available datasets for re-ID in animals. However, all available datasets are image-based, meaning that each sample is a single image showing a specific animal. To be able to develop a video-based re-ID method, the need for a video-based dataset is a matter of urgency.

To the best of our knowledge, we introduce the first video-based dataset for animal re-ID to date: the *PolarBearVidID* dataset. It includes 13 individual polar bears, each with at least 100 annotated sequences incorporating the movement of the animals. The main concept of the dataset is depicted in Figure 1. Polar bears are particularly challenging as individuals lack prominent distinct visual features. Two biologists independently labeled the data in a competitive procedure aimed at ensuring a high quality of the annotations. Furthermore, we present a novel approach to extend labels to multiple adjacent frames and reduce the labeling effort for video-based datasets. The *PolarBearVidID* dataset can be used for research and development of algorithms and systems for video-based animal re-identification. It is the first dataset that allows for the investigation of using the movement of individual animals as a feature for the task of re-ID. As this dataset directly contributes to the improvement of automated animal behavior analysis systems, it may have applications in fields such as wildlife conservation and animal behavior research.

To benchmark the *PolarBearVidID* dataset, we chose a state-of-the-art video-based method introduced by Li et al. [29] for human re-ID. It is sensitive to the movement characteristics of the individual being classified, making it ideal for benchmarking the dataset. We compare the performance to an image-based method to estimate the advantages of a video-based re-ID approach.

In summary, with *PolarBearVidID*, we contribute the first-ever video-based dataset for animal re-identification similar to the benchmark datasets for humans. This dataset allows us to test a state-of-the-art video-based approach for re-ID in animals for the very first time. Finally, we compare this method to an image-based baseline. This allows us to investigate whether the use of temporal information of the movement yields performance improvements.

## 2. The PolarBearVidID Dataset

With a sequence length of up to eight seconds, a frame rate of 12.5 fps and at least 100 sequences per animal, the *PolarBearVidID* dataset contains 138,363 identity-annotated images. As it is not feasible to manually annotate this many images, we developed a pipeline for the generation of sequences for each annotated image. This reduces labeling effort to 1%, as only one frame per sequence (consisting of up to 100 frames) needs to be annotated. In the following, we describe the data collection phase, the labeling process and how a video sequence is generated for each of the labeled images.

### 2.1. Data Collection

We collected videos in six different zoos, each housing at least two polar bears. An overview of the participating animals and institutions is depicted in Table 2. One challenge in our setting is that the animals’ enclosures are designed to be very diverse. In order to ensure consistent data recordings, we filmed the animals in up to three different suitable camera angles. The cameras cover as large an area of the enclosure as possible, which is also the natural setup commonly used in automated animal observation. We conducted the recordings using a tripod and consistently filmed with at least 12.5 frames per second.

### 2.2. Labeling Process

In order to train and evaluate a deep learning model, the identity of the animals in every single image must be determined first. Unfortunately, manual labeling is very time and labor-intensive and unfeasible in our case, as the dataset includes over 130,000 images. Furthermore, it is especially challenging for polar bears because only experts can distinguish this species reliably. In the first step, single frames are extracted from the videos, and two biologists create a bounding box for each visible animal. In the second step, they assign them to a polar bear identity. An overview of this process is depicted in Figure 2. This process is performed competitively, which means that the two experts do not know what the other person has decided. The instances of the dataset for which the identity was not consistently determined are discussed with both experts. Depending on whether they were able to agree on a common statement, the image was either kept or removed. This procedure excluded 10 images.

### 2.3. Sequence Generation

The annotated frames form the basis for further processing. We generated an eight-second video sequence for each individual frame, starting four seconds prior to the annotated frame and ending four seconds after the selected frame. The resulting sequences feature a frame rate of 12.5 fps, including a total of 100 frames. We label the remaining images in this short video sequence using the following procedure. First, we apply a state-of-the-art object detection algorithm [36], which has been trained on the class *polar bear* to each frame of an unlabeled video sequence. The output of this process should be the identification and localization of any polar bear instances present in the video frames. The object detection algorithm will analyze each frame and output the locations and bounding boxes of any polar bear instances detected within the frame. Of course, it is possible that more than one bear would be found in an image. To ensure that the correct individual is being tracked, we calculate the distance of the bounding boxes between two consecutive images. We map the boxes with the smallest distance to the same individual, as we assume this to be a correct match. The procedure is shown in Figure 3. Each individual track of one individual found during this process is cropped from the original video and saved. In order to exclude errors, we checked all sequences manually and shortened them if necessary.

### 2.4. Dataset Statistics

*PolarBearVidID* includes 13 individual polar bears housed in six institutions. Each identity has 110 sequences on average. The maximum length of the sequences is 8 s, correspondingly, 100 frames at a frame rate of 12.5 frames per second. The average length of the sequences is 96.69 images. In total, the dataset includes 1431 sequences. The resolution of the images is set to 256 × 128 pixels. Finally, *PolarBearVidID* is the first dataset to enable utilizing the movement of a non-human species as a feature for the task of re-identification. We provide the data including all relevant annotations under public license (see Data Availability Statement).

## 3. Methodology

This section provides a detailed description of both the image- and video-based re-ID methods. Furthermore, we present the training and testing procedures as well as the metrics used for evaluation. Both methods have been trained and evaluated as comparable as possible. However, there is an important distinction regarding the terminology: for the image-based approach, a *data sample* is a *single frame*, whereas for the video-based approach, a *data sample* means an entire *sequence*.

Both models follow the same processing pipeline. The input is one data sample depicting an individual polar bear to be re-identified (the *query*). The model outputs a feature vector (i.e., a mathematical representation) corresponding to that query sample. Meanwhile, both models need to have a *gallery* including examples of all individuals. All entries in the gallery are embedded into the feature space using the same procedure used for the query sample processing. Now, we compare the feature vector of the query sample using the Euclidean distance metric to every feature embedding in the *gallery* to receive the ranked list of individuals. The closer an individual from the gallery is to the query, the more likely it is that it resembles the same identity. This ranked list is the output of both re-ID approaches. When used in an automated observation system, the first rank in this list would be determined to be the identity.

### 3.1. Video-Based Benchmark

To create a video-based re-ID benchmark, we chose the model *Global-Local Temporal Representations For Video Person Re-Identification* (GLTR) introduced by Li et al. [29]. This approach was designed to utilize both spatial as well as temporal information embedded in video-based datasets. The model’s compounds are two sub-networks. The first is a backbone ResNet50 that extracts a feature vector for each frame of a video sequence analogue to the image-based baseline approach. The second sub-network inputs the frame feature vectors from the backbone and combines them into a single feature vector for the entire video sequence. That same second network is designed to model the short-term temporal cues between adjacent frames and capture the long-term relationships between frames that are further apart. The short- and long-term temporal features are aggregated with a final simple single-stream CNN [29].

### 3.2. Image-Based Baseline

We require an image-based baseline to estimate the improvement video-based approaches can provide for re-ID in animals. For this, we trained and tested a straightforward image-based method. Basically, we kept the approach as similar as possible to the video-based method, leaving out the temporal cue utilization for re-identifying. Comparable to the video-based approach, which utilizes a ResNet50 [37] as its backbone, we also use this CNN architecture for the image-based implementation. Furthermore, we developed and evaluated an alternative version of the image-based method to address the limitation that, unlike GLTR, the approach defined here inputs a single input frame. For the sake of the readability of this work, we present this in Appendix B.

### 3.3. Normalization

As we acquired the data with different cameras and at different enclosures, the videos show varying lighting conditions and color calibrations. Therefore, we normalized the images before training and testing to reduce a possible bias. For normalization, we calculated the mean and standard deviation for each color channel throughout the whole data.

### 3.4. Training Procedure

We kept the training procedure the same for both the image- and the video-based approach. This ensures the comparability of both methods. First, each training sample is processed into a feature vector. This feature vector is mapped to a single class by an additional classification layer. This single class corresponds to the individual’s ID shown in the dataset sample. Figure 4 shows the training procedure. Note that, for the image-based method, the instances of the dataset are single images, whereas, for the video-based approach, one instance corresponds to one sequence.

### 3.5. Evaluation Procedure

For the evaluation procedure, we compute the embedding vector for every instance of the test set. All test samples end up in the *gallery*. For the evaluation, we take out one entry of the *gallery* and use it as the *query*. Then, we compare the *query* to the embeddings of the gallery embedding using the Euclidean distance. This procedure results in an ascending ordered list, meaning entries in the gallery, which are deemed similar to the query sample, are at the top of the list. This ranked list of individuals is further used in the evaluation to calculate all relevant metrics (see Section 3.6). The overall procedure is depicted in Figure 4. Note that we only used one image per sequence for the image-based dataset.

### 3.6. Metrics

The use case in zoological institutions is equivalent to a closed-world setting, meaning that all individuals to be identified are known. Hence, the re-ID approaches compare a query image or video of an individual to be identified with the instances of known animals stored in the gallery to retrieve an ascending list of gallery images or videos with the most similarities to the given query. A commonly used metric to measure the performance of re-ID methods is the rank-k accuracy. It describes the probability that a correct example from the gallery with the same ID as the query is in the first k elements of the resulting list [11,38,39,40]. Naturally, in many applications, especially in zoos, only the rank-1 metric is of interest, as we need to identify the correct individual to conduct further behavioral analysis.

Another helpful metric is the mean average precision (mAP) [41,42]. It describes the average recognition performance compared to the best theoretically possible re-ID method. The metric calculation is based on the precision value for each *k*-rank, and a subsequent averaging step. The mAP is a good measure to determine how well a re-ID solution can re-identify people in a given database.

## 4. Experiments

In this section, we describe the evaluation of both the image-based and the video-based re-ID model for the *PolarBearVidID* dataset. We performed the same experiment for each approach, keeping all parameters and procedures as similar as possible.

### 4.1. Cross-Validation

As the polar bear dataset is rather small, variations in the train and test data can have a larger impact on the performance of the system. We address this issue by performing a five-fold cross-validation when training the models. First, we split the dataset into five equally sized parts. Then, we train the algorithms on four parts and evaluate the performance on the unseen fifth part. This allows us to make a more general statement about the generalization capabilities of the models. For all described experiments, we used the same data splitting according to Table A1.

### 4.2. Results

We assessed the performance of two approaches to re-identify the individuals included in the *PolarBearVidID*. Table 3 shows the results for each the image- and video-based method. The rank-1 score, as well as the mean average precision (mAP), are given as a mean result of all runs of the five-fold cross-validation, including the overall standard deviation. The data distribution over the five folds was kept the same for both approaches.

To determine which misidentifications occurred, we show the confusion matrix for both approaches in Figure 5. We display the sum over all folds. With the predicted labels (equivalent to the rank-1 instances) compared to the ground truth, the matrices display how many sequences from each individual polar bear are correctly or falsely identified. Ideally, the model outputs the correct identity, which is shown as the entries on the diagonal of the matrix. We sorted the identities according to the corresponding zoos, also denoted in the matrices. Therefore, one can assess whether misidentifications occur inter- or intra-zoos.

To assess the final use case, we evaluated the trained video-based model on the individual zoos. Thus, only the animals from the respective institution are in the Gallery. The distribution of the 5 folds remains the same as in the previous experiments. The results are shown in Table 4.

## 5. Discussion

The *PolarBearVidID* dataset is the first video-based re-ID dataset for animals. It offers many possibilities and comes with some limitations, which we will discuss in the following. Furthermore, we evaluated the performance of a state-of-the-art video-based re-ID approach and compared it to an image-based method.

### 5.1. Dataset

With the *PolarBearVidID* dataset, we provide the first-ever video-based dataset for the task of re-ID in animals. For this dataset, we present the first implementation of a video-based model utilizing state-of-the-art re-identification techniques developed for human subjects. While this evaluation is limited in scope, it serves as an initial exploration of the feasibility of transferring re-identification techniques developed for human subjects to the animal domain. By choosing polar bears, we introduced a particular challenge to the dataset, as this species lacks prominent distinct visual features and is, therefore, more difficult when it comes to re-ID.

Another specific challenge is the relatively unconstrained zoo setting. In a laboratory setting, data acquisition is performed in a controlled environment with a fixed camera angle, high camera resolution, consistent lighting conditions, a small enclosure, a uniform background and minimal occlusion of the animals. These constraints allow for more uniform data collection. However, in the zoo setting, all these parameters may vary and be less controlled. This applies intra-zoo when, for example, the weather changes or the topology of the enclosure requires cameras at different heights. Inter-zoos, of course, the parameters change even more drastically due to different enclosure designs. Therefore, data acquisition in different zoos results in a rather unconstrained re-ID dataset compared to one created in a lab. However, a re-ID method developed on a dataset recorded under laboratory conditions is not necessarily suitable for the more open zoo setting. Therefore, the *PolarBearVidID* dataset more closely reflects real-world conditions and is more representative of realistic re-ID scenarios.

One reason why all animal re-ID datasets to date are only image-based might be that the annotation of videos is a big challenge. In the case of our *PolarBearVidID* dataset, each sequence contains up to 100 individual images. Only the method of extrapolation presented by us using an object detection algorithm allows annotating this amount of images. With this method, it will be easier to create more video-based datasets for animal re-ID in the future. Note that for other species that live in herds or move close together, one might have to use a more advanced tracking method (e.g., [43]). In the end, although the experts only had to annotate 1431 images, *PolarBearVidID* offers a large number of images with almost 140,000 images compared to other publicly available datasets (see Table 1).

Finally, many authors of public datasets do not provide information about the quality of ground truth annotations or details about how they were created (e.g., [24,27,28]). Our scope was to ensure that the annotations of the *PolarBearVidID* dataset provide the highest possible quality. Especially the identification of individuals of a species such as polar bears lacking distinct coat features is very challenging and can only be performed by experts. Therefore, each data sample was annotated independently by two biologists. Unclear identities were discussed in a second step. The samples for which no agreement could be reached were excluded from the dataset. The fact that only 10 sequences had to be removed during this process shows that the level of agreement between the experts was very high. Overall, the *PolarBearVidID* dataset can be assumed to comprise labels of excellent quality.

The main limitation of our dataset is the rather small number of individual animals. In contrast, video-based benchmark datasets for humans usually include hundreds of individuals (e.g., *MARS* [11]), recorded only by a few cameras that do not change position (for example, at an intersection). However, the recording of animals in zoos is restricted to a significantly smaller number of individuals in a single enclosure. This is due to species-specific requirements and is usually limited to a maximum of three animals in the case of polar bears. This key difference limits a fair comparison between the domain of human re-ID and the domain of polar bear re-ID. In particular, the setting for human benchmark datasets cannot be replicated, and, therefore, all future video-based re-ID datasets for animals will differ substantially from those for humans, including the *PolarBearVidID* dataset.

### 5.2. Re-Identification Performance

We find that the video-based approach outperforms the image-based method in both scorings. While the image-based model attains a rank-1 score of 85.3±1.1%, the video-based one achieves an impressive score of 96.6±1.6%. Since the rank-k metric for k = 1 indicates how often the exact identity is determined for a query sample, the video-based model is perfectly suited for application in the zoo setting. The small standard deviation of the scores over the five folds indicates that both models perform very robustly on the dataset and that the train/test distribution does not influence the scores. Li et al., who introduced the video-based method (GLTR) in 2019, reported a performance of the approach on the benchmark dataset MARS with a rank-1 score of 87.02%. Thus, GLTR performs slightly better on the *PolarBearVidID* dataset. However, this statement is subject to the limitation that with 1261 individuals and over 20,000 sequences, the quantity of MARS is not comparable to our dataset.

The mean average precision of the two models differs significantly. While the video-based approach shows an excellent mAP of 88.2±9.0%, the image-based approach only achieves an mAP of 45.4±2.4%. This metric allows a good insight into the robustness of a method since it not only considers the first entry of the ranked list but also considers the remaining entries with descending weighting. The image-based method outputs the correct identity on the first rank relatively often. However, at the same time, it ranks other gallery samples of the correct identity quite often far down in the list. This means that this approach only attains a satisfactory rank-1 score because in the *PolarBearVidID* dataset for each query sample, there is often a very similar sample in the gallery. If we were to shrink the gallery size or record a more diverse dataset in the wild, the image-based method would no longer work properly. The video-based method promises much better robustness here.

The confusion matrices depicted in Figure 5 provide insight into the mismatches between predicted ID and actual ID. The matrices show that the image-based approach determines the wrong identity much more frequently. In total, the prediction does not match the actual label 211 times over all 5 folds of the test set, whereas this number drops to only 49 false identities for the video-based approach. It shows that the inter-zoo confusions almost vanish, from 81 false predictions to only 6 for the video-based model. Here, 89.58% of all misclassifications occurred within the same zoo, while only 10.42% identities were mistakenly matched to one of another zoo. This observation is to be expected, as the recordings of the animals in the same institution are identical in camera perspective and background. The zoos differ slightly in the occurrence of this issue, with inter-zoo misclassifications occurring more frequently in Berlin and Vienna. Overall, however, this finding is considered positive since, within one zoo, the animals can still be distinguished with sufficient accuracy. The influence of camera position and background is, therefore, not problematic.

This work’s use case is the re-ID method’s application in zoological institutions. When the model is deployed, the gallery includes only the animals kept in the respective enclosure. For the participating zoos, this means two or three animals, respectively. The results of this experiment are shown in Table 4. The important rank-1 score is above 90% for all zoos, meaning that wrongly classified identities occur in less than 10% of all attempts. The animals in Berlin are most frequently misidentified. The rank-1 score is 92.0%. One possible explanation is that the two polar bears in Berlin are mother and daughter. Biologists also reported that distinguishing between the two animals was particularly challenging. For this zoo, the mAP score is also the smallest, at 84.7%. This is due to the fact that in the ranked list, a false identity is more often listed in the top ranks. However, the performance for all zoos is very promising and within the scope of this project. With a re-ID performance of >90%, false detections can easily be corrected by common interpolation or filtering methods.

One limitation in evaluating both methods is the composition of the gallery. Due to the limited data set size, we used each sequence from the test set once as a query, while all remaining sequences constitute the gallery set. This procedure has a positive influence on the rank-1 scores. For the models to place the correct identity at the top of the list, there needs to be a proximate identity in the feature embedding space. A more extensive gallery helps in this regard. For the image-based approach, a smaller gallery will quickly become a problem, as suggested by the poor mAP. The video-based approach, however, has a very high mAP score, so the limitation here is less severe.

Finally, it can be concluded that the video-based approach, which utilizes the movement of the animals as a feature for re-ID, achieves a significant improvement for this task. As a result, these methods will become the go-to solution for open settings such as zoos or the wild in the coming years.

## 6. Conclusions

The greatest challenge and limitation within the animal re-ID research area is the availability of data [20]. With the *PolarBearVidID* dataset, we contribute not only a novel re-ID dataset for a species not published before but also the first-ever dataset to include fully labeled sequences of the individual animals. Furthermore, with our novel sequence generation procedure that reduces annotation effort drastically, we hope to encourage other researchers to contribute further datasets for the task of video-based re-ID. Only if the number of datasets for other species keeps increasing can the previous limitations be overcome and the field of re-ID for animals be further developed.

Developing re-ID models will greatly facilitate the work of biologists and animal caretakers in the future. Progress in this area of research enables the deployment of automated behavior observation systems and therefore contributes to the evaluation of animal welfare. With improved observation methods, the focus of biologists and veterinarians can be shifted to early recognition of behavioral changes as signs of disease, stress situations within groups, as well as the effectiveness of changed management procedures or the use of enrichment items [2,44,45].

While the *PolarBearVidID* is still limited in size, the success of our dataset and the video-based model shows that utilizing an animal’s movement as an indicator of its identity will become even more critical when we shift the scenario to the wild. In order to study animal welfare in the wild, conserve animal habitats, protect animals and preserve biodiversity, animal populations and movement patterns have to be detected and evaluated. This is performed with the help of photo traps [46]. However, with classical image-based approaches, it is not yet possible to determine single individuals on images of low quality. Therefore, animal observation and welfare investigations will significantly benefit from future developments in video-based approaches for animal re-ID. Therefore, we want to encourage the research community focused on re-ID in animals to use the *PolarBearVidID* dataset to drive further investigation in video-based methods for this task.

## Figures and Tables

**Figure 1 animals-13-00801-f001:**
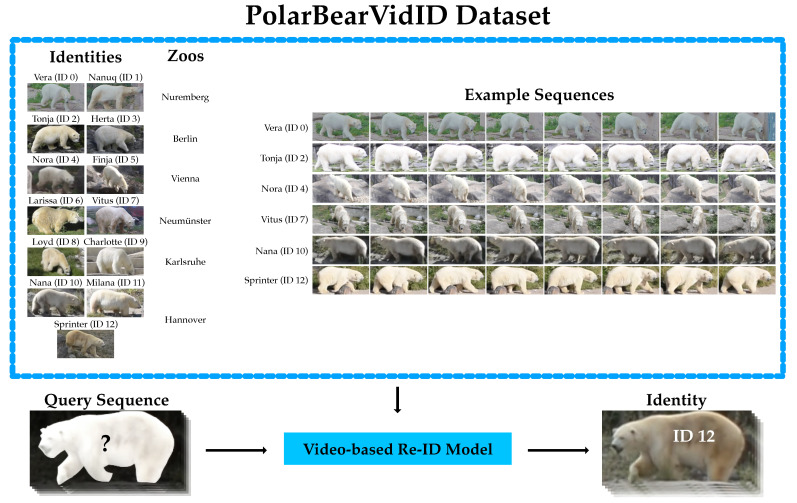
The *PolarBearVidID* dataset. It is the first dataset to date to include the movement of non-human animals to utilize this as a feature for re-ID. *PolarBearVidID* includes 13 individual polar bears housed in six institutions. Each identity has at least 100 sequences. The maximum length of the sequences is 8 s, respectively 100 frames at a frame rate of 12.5 frames per second. In total, the dataset includes 1431 sequences.

**Figure 2 animals-13-00801-f002:**
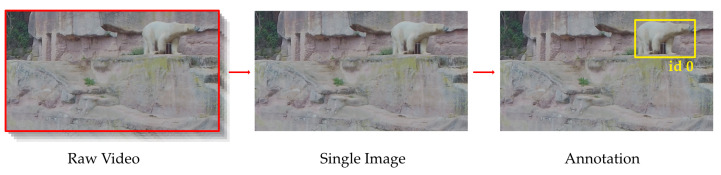
First, single frames are captured from the videos recorded in the zoos. Next, two biologists create bounding boxes for the animals and label their identities. They do not know the other person’s decision during the labeling process. The agreement between the two experts is checked across the whole dataset. Those instances in which they disagreed regarding the identity of the shown animals were collaboratively discussed. Only ten instances remained unclear. We excluded those from the dataset to ensure the highest quality of labels.

**Figure 3 animals-13-00801-f003:**
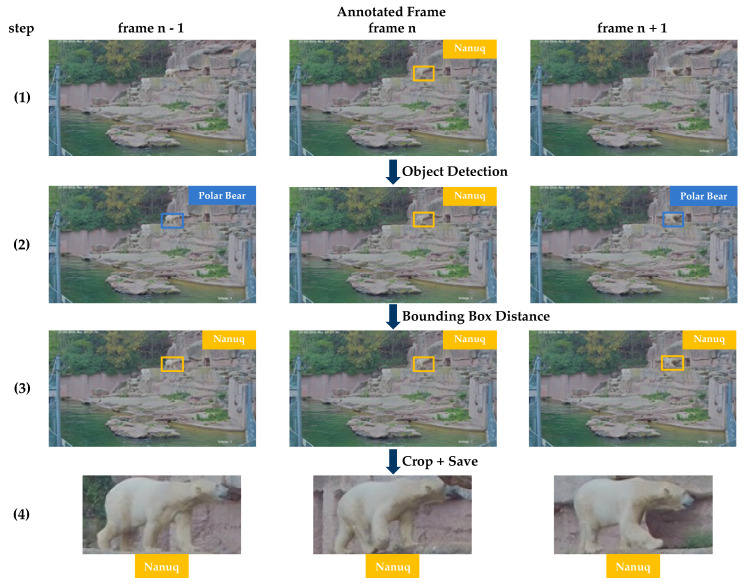
Approach to creating the sequences. For each image labeled by the experts, there is a bounding box including information about the identity of the animal (step 1). The adjacent images are then searched for polar bears in the second step, using an object detection algorithm. To ensure that the correct animal is tracked, we compute the bounding box distances between two consecutive images. If this is below a threshold, we can assign the same ID to the found animal (step 3). In step 4, the bounding boxes are then cropped and saved as a sequence.

**Figure 4 animals-13-00801-f004:**
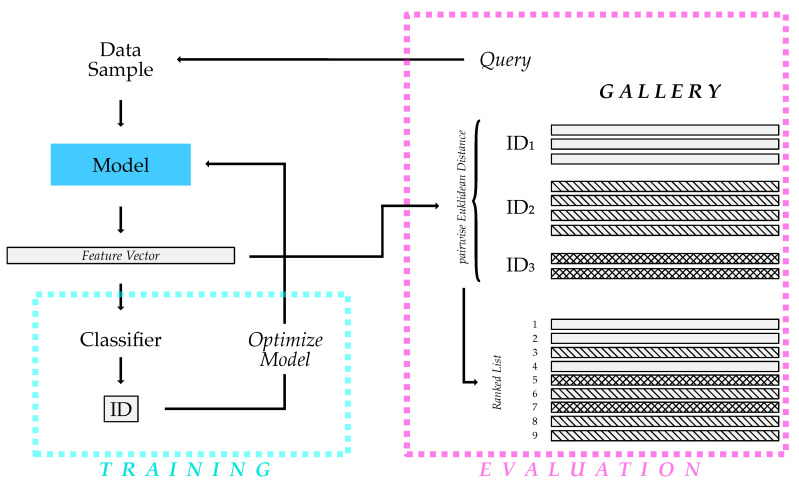
Schematic depiction of the training and testing procedure for each approach. We trained each model using a simple classifying layer. To evaluate the performance, one data sample from the test set is used as the input for the model. We calculate the Euclidean distance of the resulting feature vector to all vectors included in the gallery. This results in a ranked list of the individuals. With this list, the rank-k accuracy and the mean average precision (mAP) can be calculated.

**Figure 5 animals-13-00801-f005:**
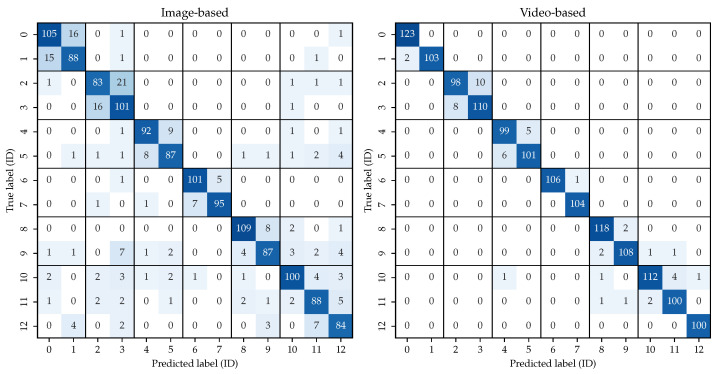
Confusion matrix of the two re-identification approaches. When confronted with a data sample of identity *X*, the model outputs a ranked list of all known identities with descending probability. Therefore, the identity *Y* listed first is the prediction of the model. In the case of X=Y, the model outputs the correct identity, which is shown as the entries on the diagonal of the matrix. The identities are sorted considering the respective zoos: Nuremberg (0 and 1), Berlin (2 and 3), Vienna (4 and 5), Neumünster (6 and 7), Karlsruhe (8 and 9) and Hannover (10, 11 and 12).

**Table 1 animals-13-00801-t001:** Publicly available datasets for the task of re-ID in animals. The *PolarBearVidID* dataset is the only one providing video sequences of the animals. Therefore, this dataset is the only one to make is possible to utilize the individuals’ movement as a unique feature for re-ID.

Dataset	Species	# Individuals	# Images	Video-Based
ATWR [22]	Amur Tigers	92	4434	✗
Chimpface [23]	Chimpanzees	90	5559	✗
ELPephants [24]	Elephants	276	2078	✗
GoldenMonkeyFace [23]	Golden Monkeys	49	1450	✗
Howard et al. [25]	Humpback Whales	4251	9850	✗
LemurFace [23]	Lemurs	129	3000	✗
Schneider et al. [26]	Fruit Flies	20	16,320	✗
SealID [27]	Saimaa Ringed seals	57	2080	✗
SeaTurtleID [28]	Sea Turtles	400	7774	✗
**PolarBearVidID**	Polar Bears	13	138,363	✓

**Table 2 animals-13-00801-t002:** Full list of the individual animals in the *PolarBearVidID* dataset.

ID	Name	Gender	Birth	Zoo	Website
0	Vera	Female	2002	Nuremberg	[30]
1	Nanuq	Male	2007	Nuremberg	[30]
2	Tonja	Female	2009	Berlin	[31]
3	Hertha	Female	2018	Berlin	[31]
4	Nora	Female	2013	Vienna	[32]
5	Finja	Female	2019	Vienna	[32]
6	Larissa	Female	1990	Neumünster	[33]
7	Vitus	Male	2000	Neumünster	[33]
8	Lloyd	Male	2000	Karlsruhe	[34]
9	Charlotte	Female	2014	Karlsruhe	[34]
10	Nana	Female	2019	Hannover	[35]
11	Milana	Female	2009	Hannover	[35]
12	Sprinter	Male	2007	Hannover	[35]

**Table 3 animals-13-00801-t003:** Performance of the image-based and video-based re-ID approach. The rank-1 score, as well as the mean average precision (mAP), is given as a mean result of all runs of the five-fold cross-validation, including the overall standard deviation.

Method	Rank-1	mAP
Image-based	0.853 ± 0.011	0.454 ± 0.024
Video-based	**0.966 ± 0.016**	**0.882 ± 0.090**

**Table 4 animals-13-00801-t004:** Performance of the video-based re-ID approach on each zoo. The rank-1 score, as well as the mean average precision (mAP), is given as a mean result of all runs of the five-fold cross-validation, including the overall standard deviation.

Zoo	Rank-1	mAP
Nuremberg	0.991 ± 0.011	0.978 ± 0.013
Berlin	0.920 ± 0.038	0.847 ± 0.070
Vienna	0.948 ± 0.041	0.854 ± 0.060
Neumünster	0.995 ± 0.010	0.976 ± 0.022
Karlsruhe	0.983 ± 0.025	0.937 ± 0.100
Hannover	0.972 ± 0.034	0.856 ± 0.159

## Data Availability

We provide the data presented in this study openly available as the *PolarBearVidID* Dataset at https://doi.org/10.5281/zenodo.7564528.

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
