# Peer review of "PolarBearVidID: A Video-Based Re-Identification Benchmark Dataset for Polar Bears"

_animals, 2023, doi:10.3390/ani13050801_

Round 1

Reviewer 1 Report

The authors provided an annotated dataset including 13 individual polar bears shown in 1431 sequences, which is an equivalent of 138,363 images. PolarBearVidID is the video-based reidentification dataset for a non-human species to date. The results show that the animals can be identified with a rank-1 accuracy of 96.6 %. It suits to this journal. Some comments are provided to improve the quality of this manuscript.

1. The main novelty of this manuscript needs to be highlighted.

2. As known, convolutional neural networks have been widely applied in many areas. What are the main differences between the used method with multi-image and the multiview CNN, e.g., “Multiview Wasserstein generative adversarial network for imbalanced pearl classification. Measurement Science and Technology, 2022, Vol. 33(8): 085406.” and “Multiview generative adversarial network and its application in pearl classification. IEEE Transactions on Industrial Electronics, 2019, Vol. 66(10): 8244-8252.”? More discussion is necessary.

3. If possible, more performance evaluation indices are needed.

Author Response

Dear Reviewer,

We thank you for the valuable feedback on our manuscript “PolarBearVidID: A Video- based Re-Identification Benchmark Dataset for Polar Bears”. We believe that it helped to improve the quality of the paper.

Please find attached detailed answers to your comments.

Best regards

Matthias Zürl

Reviewer 2 Report

The manuscript presents a dataset for polar bear re-identification and benchmark results using a video-based and an image-based method. The manuscript is well-written and interesting to read, and Introduction provides a nice overview of the field. The justification for the publication of the presented dataset is clear and sound. The dataset is properly described and can potentially be valuable to the research field. The experiments and results are clearly presented and the discussion of the results is informative. There could be more than one method or multiple settings for the video-based approach, but in my opinion overall the manuscript is ready for publication. 

Author Response

Dear Reviewer,

We thank you for the valuable feedback on our manuscript “PolarBearVidID: A Video- based Re-Identification Benchmark Dataset for Polar Bears”. We believe that it helped to improve the quality of the paper.

Please find attached detailed answers to your comments.

Best regards,
Matthias Zürl

Reviewer 3 Report

The authors present an interesting manuscript on video-based dataset for animal re-identification. Study design and research questions are clearly described. In this sense, it is easy to understand the aim of this study. The bright side of the manuscript is that to provide some useful practical details on related topic. In this context, the study contributes to different fields. Only minor concerns were raised, and manuscript needs minor structural changes. Therefore, I would like to make some suggestions to improve the quality of the paper as below:

Lines 38-39: “The development of camera-based observation systems has increasingly been carried out in the last few years as advances in deep learning continue to offer new possibilities” -> “The development of camera-based observation systems and as advances in deep learning apllications offer new tools to study animal ecology and behaviour (please add these references: 10.1126/sciadv.aaw0736 and 10.3390/ani10071207)” In my opinion such sentence would make the bridge between the problem and objectives of the study stronger.

Lines 111-114: “In summary, with PolarBearVidID we contribute the first ever video-based dataset for animal re-identification similar to the benchmark datasets for humans. We evaluate a state-of-the-art video-based approach utilizing temporal information for the task of re-identification. Finally, we compare this method to an image-based baseline.” I think, this paragraph should moved to conclusion section.

Lines 115-170: “2. The PolarBearVidID Dataset” should be a subsection of Materials and Methods section since the authors defines the materials of the study. In this context; 2. The PolarBearVidID Dataset -> 2.1. 2. The PolarBearVidID Dataset

Line 170: 2. Methodology -> Materials and Methods. Please see Journal’s Instructions for Authors: https://www.mdpi.com/journal/animals/instructions

Lines 248-259: “4. Experiments” should be a subsection of Materials and Methods section since the authors defines the experimental method of the study.

Line 260: “4.2 Results -> 3. Results

Line 277: 5. Discussion -> 4. Discussion

Line 399: 6. Conclusion -> 5. Conclusion

Author Response

(The authors gave the same response as above.)
